# SELF-GUIDED PROCESS REWARD OPTIMIZATION WITH REDEFINED STEP-WISE ADVANTAGE FOR PROCESS REINFORCEMENT LEARNING

## ABSTRACT

Process Reinforcement Learning (PRL) has demonstrated considerable potential in enhancing the reasoning capabilities of Large Language Models (LLMs). However, introducing additional process reward models incurs substantial computational overhead, and there is no unified theoretical framework for process-level advantage estimation. To bridge this gap, we propose **S**elf-Guided **P**rocess **R**eward **O**ptimization (**SPRO**), a novel framework that enables process-aware RL through two key innovations: (1) we show that process rewards can be derived intrinsically from the policy model itself, and (2) we redefine the step-wise advantage by introducing well-defined Cumulative Process Rewards (**CPR**) and **M**asked **S**tep **A**dvantage (**MSA**), which facilitates rigorous step-wise action advantage estimation within shared-prompt sampling groups. Our experimental results demonstrate that SPRO outperforms vanilla GRPO with 3.4× higher training efficiency and a 17.5% test accuracy improvement. Furthermore, SPRO maintains a stable and elevated policy entropy throughout training while achieving a considerable reduction in the average response length, evidencing sufficient exploration and prevention of reward hacking. Notably, SPRO incurs no additional computational overhead compared to outcome-supervised RL methods such as GRPO, which benefit industrial implementation.

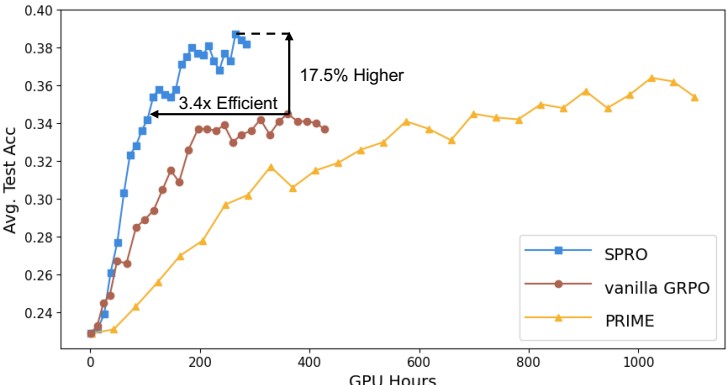

Figure 1: **Performance comparison on math and code benchmarks up to 432 steps.** SPRO outperforms outcome-supervised GRPO with 3.4× higher training efficiency and a 17.5% test accuracy improvement. Notably, SPRO reduces per-step computation time owing to its shorter trajectories.

## 1 INTRODUCTION

Reasoning ability is fundamental to the intelligence of language models and plays a pivotal role in advancing artificial general intelligence (AGI). Recent open source reasoning models, such as OpenAI's o1 (OpenAI, 2024) and DeepSeek's R1 (Guo et al., 2025), demonstrate the effectiveness

of reinforcement learning in reasoning tasks. Most current RL algorithms (Guo et al., 2025; Shao et al., 2024; Team et al., 2025; Ahmadian et al., 2024) optimize policy models based exclusively on outcome rewards, neglecting intermediate feedback. This sparse reward paradigm leads to inefficient learning (Qu et al., 2025; Cui et al., 2025b), highlighting the need for efficient and scalable process-based reinforcement learning algorithms. This raises a fundamental research question: *how to provide well-defined process rewards and guide the policy optimization effectively?*

Although significant research (Lightman et al., 2023; Feng et al., 2023; Snell et al., 2024; Wang et al., 2024) has been devoted to addressing this question, there is no unified theoretical framework for process-level advantage estimation. Training auxiliary process reward models (PRMs), which estimate the future success of intermediate steps, is a commonly adopted strategy. However, PRMs exhibit several widely recognized practical limitations:

**(1) Difficult to train:** Human-annotated process-level labels lack scalability (Lightman et al., 2023), while automatic annotation often fails to provide reliable supervision (Guo et al., 2025).

**(2) High computational cost:** In contrast to the widely adopted dual-model framework (policy and reference model) in outcome-supervised algorithms such as Guo et al. (2025), Shao et al. (2024) and Ahmadian et al. (2024), which significantly contributes to scalability and industrial adoption, PRM-based methods introduce an auxiliary reward model. Loading of the additional model requires considerable GPU memory allocation, which constrains the batch size and substantially degrades training throughput and efficiency.

**(3) Non-scalable utilization:** Existing methods typically leverage PRMs to rerank candidate responses (Uesato et al., 2022) or perform Monte Carlo Tree Search (MCTS) (Lightman et al., 2023; Feng et al., 2023; Wang et al., 2024), aiming to improve reasoning trajectories. However, constructing the reasoning search space requires sequential rollouts at each step (Snell et al., 2024), making these approaches non-scalable in online RL.

Recent works have proposed novel methods for acquiring high-quality PRMs. Rafailov et al. (2024) demonstrate that a well-trained DPO model can inherently achieve credit assignment, effectively expressing token-level rewards in the Markov Decision Process (MDP) framework of LLMs. Extending DPO framework, Yuan et al. (2025) introduce a more generalized implicit PRM training paradigm that replaces preference pairs with point-wise labeled trajectories, which can be trained using cross-entropy loss. Subsequently, Cui et al. (2025a) improve the implicit PRM methodology by proposing the PRIME framework, which effectively combines token-level rewards with outcome rewards to calculate trajectory advantages. The derived token-level reward functions proposed in Rafailov et al. (2024), Yuan et al. (2025), and Cui et al. (2025a) eliminate the need for explicit process annotations, thus addressing the practical limitation (1) and streamlining the training pipeline.

However, PRIME (Cui et al., 2025a) relies on an auxiliary reward model $\pi_\varphi$ to parameterize implicit PRM, which inherits the practical limitation (2) and requires iterative training updates throughout the optimization process. This approach not only consumes additional GPU memory but also introduces non-negligible computational overhead. Additionally, although PRIME (Cui et al., 2025a) circumvents limitation (3) by estimating advantages via a Monte Carlo estimator combined with a leave-one-out baseline, it aggregates all process rewards into a single group for normalization (see Fig. 2a). Such a treatment deviates from standard advantage-based policy gradient methods like PPO (Schulman et al., 2017), thereby introducing significant estimation bias.

To address the computational inefficiency of auxiliary PRMs in industry-scale PRL while achieving a more reasonable process advantage estimation, in this paper, we propose Self-guided Process Reward Optimization (SPRO), a PRM-free algorithm for process reinforcement learning as shown in Fig. 2b. We demonstrate that process rewards can be self-guided directly from the policy model itself, as SPRO eliminates both the annotation requirements and computational overhead inherent to PRM-based approaches and preserves the simplicity and scalability of outcome-supervised RL algorithms (Shao et al., 2024; Ahmadian et al., 2024), which benefit the industrial implementation.

Moreover, SPRO offers a theoretical framework for step-level advantage estimation by redefining the step-wise advantage through a novel Cumulative Process Reward (CPR). This approach aligns with the classic advantage-based policy gradient framework by leveraging the nature of masked attention, which structurally encodes prefix-sequence information. Specifically, CPR implicitly aggregates the process rewards from all preceding steps in the prefix sequence as a surrogate for process rewards, enabling more accurate expected return estimation at each timestep. For advantage estimation, we extend the formulation of group-relative advantage from outcome-supervised algorithms (Shao et al.,

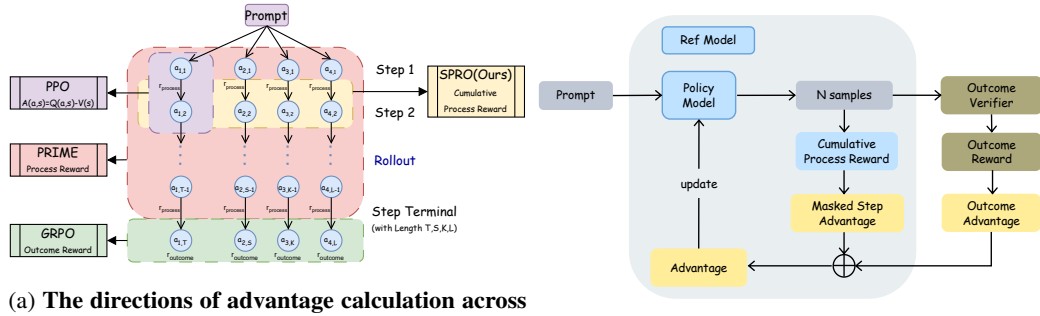

(a) **The directions of advantage calculation across different framework.**

(b) **Illustration of SPRO.**

Figure 2: (a): PPO estimates advantages based on single-step state transitions. GRPO treats all terminal states as one-step transitions from the initial prompt state. PRIME aggregates all process rewards into a single group for normalization. In contrast, we propose SPRO, which groups rewards at the same step for calculation. (b): SPRO utilizes Cumulative Process Rewards directly derived from the policy model, thereby eliminating the need for an additional process reward model and establishing a dual-model framework comparable to outcome-supervised approaches.

2024; Ahmadian et al., 2024) and investigate the estimation of step-level advantage. To enable fair comparisons, we introduce Masked Step Advantage (MSA), which enforces strict per-step comparison within shared-prompt sampling groups.

As shown in Fig. 2a, we compare advantage functions across mainstream methods. Assume that four responses are sampled, where each response receives a outcome reward, and each intermediate step is assigned a process reward. GRPO (Guo et al., 2025) estimates policy gradients by computing relative advantages within *trajectory groups* using outcome rewards, while PRIME (Cui et al., 2025a) adapts the grouping paradigm to all the token-level rewards for normalization. As for SPRO, we employ the Cumulative Process Reward (CPR) to compute the step-wise rewards at step $t$. Subsequently, for each identical step across different trajectories, group-wise normalization is applied to the rewards to obtain the Masked Step Advantage (MSA).

The experimental results demonstrate significant improvements of SPRO over baseline methods. As shown in Fig. 1, SPRO achieves $17.5\%$ higher test accuracy than vanilla GRPO and $8.3\%$ higher than PRIME, while reducing computational costs to $29\%$ (vs. GRPO) and $15\%$ (vs. PRIME) of GPU hours for equivalent performance. The comparisons on response length and policy entropy also demonstrate that our approach simultaneously addresses two long-standing challenges that have attracted significant community attention: (1) improving token efficiency in reasoning (Qu et al., 2025; Liu et al., 2025), and (2) mitigating policy entropy collapse (or reward hacking) during RL training (Cui et al., 2025b). These phenomena are discussed in Sec. 3.3. This dual improvement indicates that our framework enables the policy model to more effectively recognize the advantages of each step, resulting in both efficient reasoning and effective action space exploration.

The main contributions are summarized as follows:

- We introduce a novel RL framework for LLMs, *Self-Guided Process Reward Optimization (SPRO)*, which eliminates the need for costly PRMs and retains the same simplicity and scalability as outcome-supervised RL.
- We redefine the step-level advantage by introducing a novel *Cumulative Process Reward (CPR)* as a surrogate for self-guided process rewards and further propose *Masked Step Advantage (MSA)*, which enables a strict per-step comparison within shared-prompt sampling groups to estimate step-level advantages.
- Our experimental results demonstrate that SPRO simultaneously improves accuracy and training efficiency while resolving two critical challenges: token efficiency and policy entropy collapse. SPRO significantly reduces the length of reasoning sequences while achieving higher accuracy. Moreover, SPRO maintains higher policy entropy, promoting more efficient exploration and mitigating reward hacking.

## 2 SELF-GUIDED PROCESS REWARD OPTIMIZATION

In this section, we propose a novel PRM-free process reinforcement learning framework for token-level Markov Decision Processes (MDPs) in LLMs. Our framework uniquely enables the policy model to serve dual roles during optimization: (i) as an **Actor** module for policy improvement through reinforcement learning, and (ii) as a **Reward** module for token-level credit assignment during the generation process. Therefore, we refer to our framework as Self-Guided Process Reward Optimization.

We introduce our framework through three steps:

- Sec. 2.1: We propose the hypothesis that any LLM can provide credit assignment for token-level MDP, which is independent of the specific training objective used to train the LLM.
- Sec. 2.2: We redefine step-wise advantage by introducing Cumulative Process Reward (CPR) for token-level MDP and further propose Masked Step Advantage for process RL training.
- Sec. 2.3: We introduce the Self-Guided Process Reward Optimization (SPRO) algorithm, detailing its objective function and training procedure.

This self-guided framework has three key advantages: (i) it avoids the reward modeling bottleneck inherent in traditional RLHF pipelines; (ii) credit assignment dynamically improves in alignment with policy improvement, creating a virtuous cycle of mutual refinement; and (iii) it enables effective process reinforcement learning.

### 2.1 THE PROCESS REWARDS COULD BE SELF-GUIDED

As shown in Eq. (14), the process reward $r(\mathbf{s}_t, \mathbf{a}_t)$ is defined as the log-ratio between the probability of $\mathbf{a}_t$ under the optimal policy and the given reference policy. Rafailov et al. (2024) argue that the trained DPO model $\pi^*$ yields the best estimate of an optimal $Q$-function, since the value term $V^*(\mathbf{s})$ is reduced using Bradley-Terry preference model. Building upon this, Yuan et al. (2025) further extend the idea to Cross-Entropy (CE) loss. Cui et al. (2025a) apply CE loss to train an implicit PRM, and subsequently use the resulting process rewards to compute advantages for policy optimization.

Obviously, the accuracy of $r(\mathbf{s}_t, \mathbf{a}_t)$ directly depends on the quality of the optimal policy. Since the policy model is trained to directly approximate $\pi^*$, the near-optimal solution $\pi_{\theta_T}$ inherently provides more accurate rewards than a PRM $\pi_\varphi$ trained separately. Otherwise, the policy model itself would be inferior, contradicting its optimality hypothesis, which means that the separately trained PRM could be a better solution than our trained policy model $\pi_\theta$. This observation forms the foundation for our self-guided reward formulation.

**Proposition 1.** *Any LLM is always the optimal soft Q-functions for some reward functions in the token-level MDP (Rafailov et al., 2024), thus enabling token-level credit assignment. In particular, LLMs with stronger downstream task performance provide more accurate credit assignment.*

*Proof.* Let $\boldsymbol{\ell}(\mathbf{a}_t|\mathbf{s}_t)$ denote the output logits of a given LLM policy $\pi$ for token $\mathbf{a}_t$ conditioned on state $\mathbf{s}_t$. We define $Q$-function as a scaled version of the logits: $Q(\mathbf{s}_t, \mathbf{a}_t) = \beta\boldsymbol{\ell}(\mathbf{a}_t|\mathbf{s}_t)$. The corresponding partition function is derived by taking the log-sum-exp of logits over all possible actions $\mathbf{a} \in \mathcal{A}$ and defined as $Z(\mathbf{s}_t)$. Consequently, the optimal value function corresponding to such $Q$ is exactly $V(\mathbf{s}_t) = \beta \log Z(\mathbf{s}_t)$. This yields the following form of the policy:

$$\pi(\mathbf{a_t}|\mathbf{s}_t) = \text{softmax}(\boldsymbol{\ell}(\mathbf{a_t}|\mathbf{s_t})) = \frac{e^{\boldsymbol{\ell}(\mathbf{a_t}|\mathbf{s_t})}}{\sum_{\mathbf{a}\in\mathcal{A}} e^{\boldsymbol{\ell}(\mathbf{a}|\mathbf{s_t})}} = \frac{e^{Q(\mathbf{s}_t,\mathbf{a}_t)/\beta}}{Z(\mathbf{s}_t)} = e^{(Q(\mathbf{s}_t,\mathbf{a}_t)-V(\mathbf{s}_t))/\beta}. \quad (1)$$

Eq. (1) shows that any LLM is a soft $Q$-function for some reward function (Rafailov et al., 2024). Since any LLM is the optimal soft $Q$-functions for some reward functions in the token-level MDP, better downstream task performance implies that the corresponding reward function is more aligned with the task, which enables the LLM to perform more accurate credit assignment. Importantly, this property is independent of the specific training objective used to train the LLM. $\square$

In our framework, we utilize the policy model itself, rather than a well-trained reward model. Since both the policy and reference models are initialized from the same SFT model, the process rewards are initially zero. As training progresses and the policy model shifts away from the reference, the process rewards start contributing to the RL optimization.

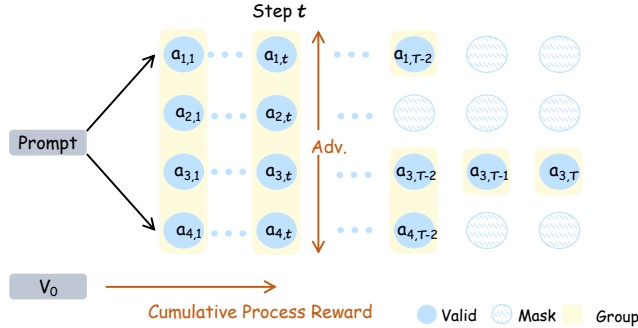

Figure 3: **Illustration of Masked Step Advantage.** Assume that four responses are sampled for each prompt. At each step $t$, we calculate cumulative process rewards and further compute the step-level advantages within the vertical valid masked groups, excluding empty step units from all calculations.

## 2.2 REDEFINE STEP-WISE ADVANTAGE

In the previous section, we propose that the process reward can be self-guided by the policy model. In this section, we redefine step-wise advantage by introducing *Cumulative Process Reward (CPR)* and *Masked Step Advantage (MSA)* to facilitate rigorous step-wise action advantage estimation.

**Cumulative Process Reward (CPR).** We argue that the mechanism of LLMs should inform the design of process rewards in token-level MDPs. Since auto-regressive generation employs masked attention, the hidden state at step $t$ inherently encodes all information of the prefix sequence (Vaswani et al., 2017), which means each hidden state represents the complete trajectory up to its corresponding time step. Prior work has effectively utilized this property: Lightman et al. (2023) employ the final token's hidden state at each step to predict correctness with PRMs, aligning with extensive probing studies that leverage such representations to analyze model properties (Belinkov, 2022; Aspillaga et al., 2021; Conneau et al., 2018; Dai et al., 2022; Geva et al., 2021; Allen-Zhu & Li, 2024). Therefore, we propose that intermediate reward signals at step $t$ should similarly capture contributions from all preceding steps, which we formalize as the definition of the Cumulative Process Reward.

Given a policy model $\pi_\theta$ during training iterations (we omit the iteration subscript for convenience), Proposition 1 establishes that there always exists an implicit reward function $r(\mathbf{s}_t, \mathbf{a}_t)$ together with corresponding $Q(\mathbf{s}_t, \mathbf{a}_t)$ and $V(\mathbf{s}_t)$ functions. Due to the optimality of the $Q$-function which is introduced in Appendix A, these functions also satisfy the equality relation in Eq.(14) as follows:

$$r(\mathbf{s}_t, \mathbf{a}_t) + V(\mathbf{s}_{t+1}) - V(\mathbf{s}_t) = \beta \log \frac{\pi_\theta(\mathbf{a}_t|\mathbf{s}_t)}{\pi_{\text{ref}}(\mathbf{a}_t|\mathbf{s}_t)}. \tag{2}$$

For an arbitrary time step $t$ within a trajectory $\tau = \{\mathbf{s}_0, \mathbf{a}_0, \dots, \mathbf{a}_{T-1}, \mathbf{s}_T\}$, we define a cumulative reward w.r.t. step $t$ by accumulating Eq. (2) from 0 to $t$:

$$\sum_{j=0}^{t} \left(r\left(\mathbf{s}_j, \mathbf{a}_j\right) + V\left(\mathbf{s}_{j+1}\right) - V\left(\mathbf{s}_j\right)\right) = \sum_{j=0}^{t} \beta \log \frac{\pi_\theta(\mathbf{a}_j|\mathbf{s}_j)}{\pi_{\text{ref}}(\mathbf{a}_j|\mathbf{s}_j)}. \tag{3}$$

By telescoping the value function $V(\mathbf{s}_{j+1}) - V(\mathbf{s}_j)$ on the left-hand side, we can get the following:

$$\sum_{j=0}^{t} r(\mathbf{s}_j, \mathbf{a}_j) + V(\mathbf{s}_{t+1}) = V(\mathbf{s}_0) + \sum_{j=0}^{t} \beta \log \frac{\pi(\mathbf{a}_j|\mathbf{s}_j)}{\pi_{\text{ref}}(\mathbf{a}_j|\mathbf{s}_j)}. \tag{4}$$

The left-hand side of Eq. (4) represents the cumulative reward up to step $t$ plus the future expected return starting from $\mathbf{s}_{t+1}$ (discount factor $\gamma = 1$). Let $\mathcal{R}_t$ denote Cumulative Process Reward (CPR):

$$\mathcal{R}_t := \sum_{j=0}^{t} r(\mathbf{s}_j, \mathbf{a}_j) + V(\mathbf{s}_{t+1}) = V(\mathbf{s}_0) + \sum_{j=0}^{t} \beta \log \frac{\pi(\mathbf{a}_j|\mathbf{s}_j)}{\pi_{\text{ref}}(\mathbf{a}_j|\mathbf{s}_j)}. \tag{5}$$

---

**Algorithm 1: S**elf-guided **P**rocess **R**eward **O**ptimization (**SPRO**)

---

**Input:** Initial policy model $\pi_{\theta_{\text{init}}}$; outcome reward verifier $r_o$; task prompts $\mathcal{D}$.

1   policy model $\pi_\theta \leftarrow \pi_{\theta_{\text{init}}}$
2   reference model $\pi_{\text{ref}} \leftarrow \pi_{\theta_{\text{init}}}$
3   **for** *iteration* $= 1$ **to** $K$ **do**
4      Sample a batch $\mathcal{D}_b$ from $\mathcal{D}$
5      Update the old policy model $\pi_{\theta_{\text{old}}} \leftarrow \pi_\theta$
6      Sample G outputs $\{\mathbf{y}_i\}_{i=1}^G \sim \pi_{\theta_{\text{old}}}(\cdot \mid \mathbf{x})$ for each $x \in \mathcal{D}_b$
7      Compute outcome rewards $\{r_o(\mathbf{y}_i)\}_{i=1}^G$ for each sampled output $\mathbf{y}_i$
8      Apply accuracy filter on prompts in $\mathcal{D}_b$
9      Compute the Cumulative Process Reward $\mathcal{R}_{i,t}$ and Masked Step Advantage $\text{MSA}_{i,t}$ for the $t^{\text{th}}$ token of all responses $\{\mathbf{y}_i\}$ with Eqs. (5-6)
10     Compute $A_{i,t}$ for the $t^{\text{th}}$ token of all responses $\{\mathbf{y}_i\}$ through Eq. (7)
11     **for** *iteration* $= 1$ **to** $\mu$ **do**
12       Update the policy model $\pi_\theta$ by maximizing the SPRO objective Eq. (8)
13     **end**
14 **end**

**Output:** Policy Model $\pi_\theta$.

---

We adopt CPR to align with the cumulative nature of LLM representations. Additionally, since all responses start from the same initial state $\mathbf{s}_0$, $\mathcal{R}_t$ facilitates subsequent advantage computation.

**Masked Step Advantage (MSA).**    For trajectories $\{\tau_i\}$ of the same prompt, the Cumulative Process Rewards $\{\mathcal{R}_{i,t}\}$ at the same step $t$ are comparable because they all start from the same initial state $\mathbf{s}_0$ and can be regarded as one-step state transition rewards, similar to GRPO that the outcome reward can be seemed as one-step transition rewards. We formally define **Masked Step Advantage (MSA)** corresponding to the cumulative reward as follows:

$$\text{MSA}_{i,t} := \mathcal{R}_{i,t} - b_t = \tilde{\mathcal{R}}_{i,t} - \tilde{b}_t = \tilde{\mathcal{R}}_{i,t} - \text{mask\_mean}(\{\tilde{\mathcal{R}}_{i,t}\}), \tag{6}$$

where $i$ represents the $i^{\text{th}}$ response and $\tilde{\mathcal{R}}_{i,t} = \sum_{j=0}^t \beta \log \frac{\pi_\theta(\mathbf{a}_j|\mathbf{s}_j)}{\pi_{\text{ref}}(\mathbf{a}_j|\mathbf{s}_j)}$. The term $b_t$ (or $\tilde{b}_t$) denotes the group-wise average of valid steps at step $t$, serving as an advantage baseline. Here, a $\texttt{masked\_mean}$ operator is employed. Note that the constant $V(\mathbf{s}_0)$ on the right-hand side of Eq. (5) will cancel out in all responses, making $\mathcal{R}_{i,t}$ and $\tilde{\mathcal{R}}_{i,t}$ equivalent for advantage calculation.

Taking Fig. 3 as an example, if only the third response contains a valid step at time $T-1$, then $b_{T-1} = \mathcal{R}_{3,T-1}$ and $\text{MSA}_{3,T-1} = 0$. This indicates that **MSA does not introduce the length bias**, since the third response does not gain additional advantage even if it is longer than the others. In this way, strict per-step comparisons within shared-prompt sampling groups are achieved without introducing length bias.

### 2.3 SELF-GUIDED PROCESS REWARD OPTIMIZATION

Following common practice in Policy Gradient algorithm (Williams, 1992), we incorporate MSA as a process reward signal into the outcome-supervised RL method Group Relative Policy Optimization (GRPO) (Shao et al., 2024), resulting in the SPRO advantage function:

$$A_{i,t} = \underbrace{\frac{r_o(\mathbf{y}_i) - \text{mean}(\{r_o(\mathbf{y}_i)\})}{\text{std}(\{r_o(\mathbf{y}_i)\})}}_{\text{GRPO with outcome rewards}} + \underbrace{\left(\mathcal{R}_{i,t} - \text{masked\_mean}(\{\mathcal{R}_{i,t}\})\right)}_{\text{MSA}_{i,t}}. \tag{7}$$

Then the policy model can be optimized by maximizing the objective as follows:

$$\mathcal{J}_{\text{SPRO}}(\theta) = \mathbb{E}_{\mathbf{x}, \{\mathbf{y}_i\}_{i=1}^G \sim \pi_{\theta_{\text{old}}}(\cdot|\mathbf{x})} \frac{1}{G} \sum_{i=1}^G \frac{1}{|\mathbf{y}_i|} \sum_{t=1}^{|\mathbf{y}_i|} \min$$
$$\left( \frac{\pi_\theta(y_{i,t} \mid \mathbf{x}, \mathbf{y}_{i,<t})}{\pi_{\theta_{\text{old}}}(y_{i,t} \mid \mathbf{x}, \mathbf{y}_{i,<t})} A_{i,t}, \text{CLIP}\left( \frac{\pi_\theta(y_{i,t} \mid \mathbf{x}, \mathbf{y}_{i,<t})}{\pi_{\theta_{\text{old}}}(y_{i,t} \mid \mathbf{x}, \mathbf{y}_{i,<t})}, 1 - \varepsilon, 1 + \varepsilon \right) A_{i,t} \right). \tag{8}$$

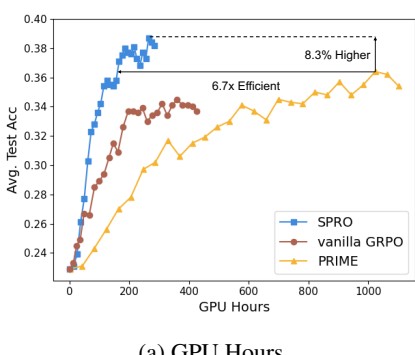
(a) GPU Hours

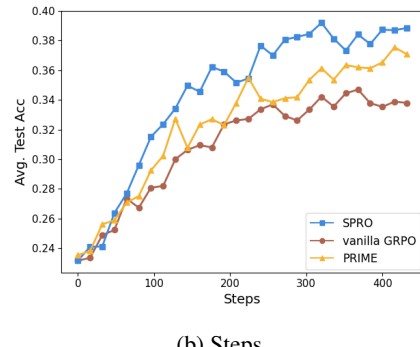
(b) Steps

Figure 4: Performance of SPRO on math and code benchmarks with respect to GPU hours and training steps. SPRO outperforms the previous SoTA process reinforcement learning method PRIME with 6.7× higher training efficiency and an 8.3% performance improvement. Notably, SPRO reduces per-step computation time owing to its shorter trajectories.

Alg. 1 illustrates the detailed implementation of our proposed SPRO framework. It can be observed that the calculations of CPR and MSA depend exclusively on the current policy model during training, which motivates our designation of the approach as self-guided. Furthermore, our proposed advantage function computes relative advantages by grouping tokens from the identical timestep across all sampled responses, ensuring a less biased advantage estimation.

## 3 EXPERIMENTS

### 3.1 EXPERIMENTAL SETUP

We conducted experiments on mathematical and programming datasets to evaluate our proposed SPRO, focusing on comparing vanilla GRPO (Shao et al., 2024) and PRIME (Cui et al., 2025a). Appendix C describes our experimental setup in detail, including **Base model and Dataset, Resources and Hyper-parameters, Outcome Verifiers, Evaluation Benchmarks and Baseline Algorithms.**

### 3.2 MAIN RESULTS

Our experiment results demonstrate that SPRO enables effective PRL for reasoning, yielding substantial improvements over GRPO and PRIME.

**Accuracy Improvement.** As shown in Fig. 1 and Fig. 4, SPRO achieves a $17.5\%$ higher test accuracy than GRPO (Shao et al., 2024), and $8.3\%$ higher than PRIME (Cui et al., 2025a). The detailed test accuracy values can be found in Table 3. The comparative evaluation results in Table 1 demonstrates that SPRO significantly enhances the reasoning capabilities of policy models.

**Token Efficiency.** Fig. 5a shows that SPRO achieves progressively shorter response lengths during training process compared to baseline methods. Table 2 presents the sampling sequence length on the validation set. Our SPRO achieves the best performance among all methods, even with a response length nearly one-third shorter than vanilla GRPO. A case study is presented in Appendix E.

**Training Efficiency.** As shown in Fig. 1 and Fig. 4a, our SPRO requires only $29\%$ and $15\%$ of the GPU hours needed by vanilla GRPO and PRIME respectively to achieve equivalent accuracy. Furthermore, the shorter sampling sequence length generated by SPRO (as illustrated in Fig. 5a) contributes to a significant computational advantage in each optimization step. This is evidenced by comparing the computation time required for the same number of training steps across different methods (see comparative results in Fig. 4).

**Entropy Stability.** As shown in Fig. 5b, unlike PRIME which suffers from entropy collapse, our method maintains effective state-action space exploration during training, preserving optimization efficiency and avoiding reward hacking.

Table 1: **Comparison of evaluation accuracy** between SPRO and other baselines under identical conditions: same base model, training data, and 432 training steps. The baselines are reproduced using the veRL codebase[1].

| Methods | AIME | AMC | MATH | Minerva | Olympiad | LeetCode | LiveCode | Avg. |
|---|---|---|---|---|---|---|---|---|
| Base Model | 3.33 | 37.35 | 64.00 | 21.69 | 28.15 | 22.22 | 16.72 | 27.64 |
| vanilla GRPO | 10.00 | 39.76 | 66.40 | 27.57 | 30.67 | 28.89 | 22.70 | 32.28 |
| PRIME | 10.00 | 43.37 | 72.20 | 28.68 | 29.04 | 28.33 | 22.16 | 33.40 |
| **Ours (SPRO)** | **13.33** | **45.78** | **74.20** | **30.15** | **39.11** | **29.44** | **24.00** | **36.57** |

Table 2: **Comparison of response length** between SPRO and other baselines under identical conditions: same base model, training data, and 432 training steps.

| Methods | AIME | AMC | MATH | Minerva | Olympiad |
|---|---|---|---|---|---|
| Base Model | 1893 | 1571 | 951 | 762 | 1206 |
| vanilla GRPO | 2028 | 1635 | 927 | 816 | 1263 |
| PRIME | 1731 | 1496 | 920 | 983 | 1323 |
| **Ours (SPRO)** | **1349** | **1024** | **685** | **701** | **947** |

## 3.3 ANALYSIS

### 3.3.1 SPRO ENABLES EFFICIENT REASONING TRAJECTORIES

Fig. 5a shows that SPRO reduces the average response length of vanilla GRPO (Shao et al., 2024) by nearly one-third while improving test accuracy by $17.5\%$, demonstrating the effectiveness of our framework in process reinforcement learning.

This improvement is primarily attributed to the rigorous step-wise comparison mechanism introduced in Eq. (6). Our approach provides the policy model with MSA feedback at each generation step, enabling it to identify which tokens contribute positively to the overall return. Such a fine-grained feedback mechanism effectively encourages more concise and task-focused output. In contrast, existing methods such as PRIME (Cui et al., 2025a) employ a coarser reward signal by averaging the returns across all trajectories and timesteps. This design results in an advantage function that is relative to both groups and timesteps, consequently diminishing the effectiveness of the intended group-level comparison.

The Cumulative Process Rewards defined in Eq. (5) not only provide the foundation for MSA in Eq. (6), but also exhibit intrinsic alignment with the hidden state dynamics of LLMs. This insight, inherent in the mechanism of LLMs and different from conventional RL scenarios, merits further attention in future research on reinforcement learning for LLMs.

### 3.3.2 SPRO ENABLES EXPLORATION OF ACTION SPACE

The collapse of policy entropy is a widely observed phenomenon in reinforcement learning, as documented in prior works (Cui et al., 2025b; Yu et al., 2025), and our experimental results are consistent with this trend. In our experiments, the entropy coefficient is fixed at $0.001$ across methods. As shown in Fig. 5, the policy entropy starts at $0.13$ due to SFT initialization. PRIME (Cui et al., 2025a) suffers a sharp entropy drop within the first 100 steps, while GRPO (Shao et al., 2024) remains largely stable. In contrast, our SPRO demonstrates active state action space exploration, whose policy entropy increases to $0.35$ and remains relatively high up to $500$ steps before gradually declining and fluctuating within a narrow range around $0.2$.

This persistent exploration constitutes a key feature of SPRO, enabling longer and more effective RL training. By maintaining output diversity, our SPRO prevents premature convergence to suboptimal behaviors while preserving the potential for further improvement. Cui et al. (2025b) demonstrate that the policy entropy naturally decreases when high-advantage actions already have high probability, but increases when the model selects rare yet high-advantage actions. SPRO encourages the latter

---

[1]https://github.com/volcengine/verl

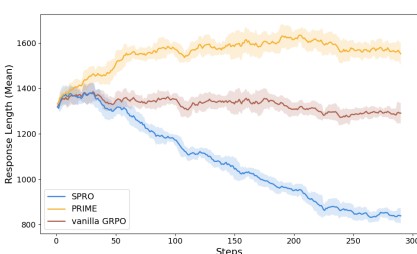

(a) Average length of policy model-generated responses.

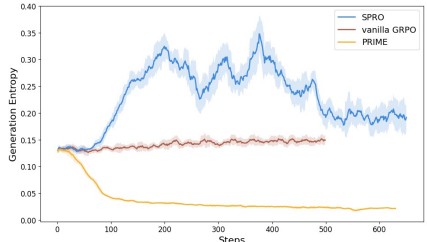

(b) Entropy of policy model's generation probabilities.

Figure 5: Effective process reinforcement learning enables efficient reasoning traces and exploration of action space.

behavior, validating the effectiveness of our advantage function design in Eq. 7. This exploration mechanism directly contributes to the $17.5\%$ improvement in test accuracy over vanilla GRPO. Importantly, this performance gain stems not from implementation tricks such as policy loss clipping, but from genuine exploration dynamics.

In particular, our SPRO successfully combines active exploration with more concise reasoning trajectories. The reduced response length does not mean shortcutting; instead, the policy model thoroughly explores the state-action space while strategically selecting concise, diverse, and effective solutions. This demonstrates a form of intelligent exploration, where the model identifies efficient solutions without compromising the correctness or diversity.

### 3.3.3 SPRO ENABLES INDUSTRY-SCALE PROCESS REINFORCEMENT LEARNING

Even in advanced industrial Large Reasoning Models, PRMs have been identified as a failure case due to their inherent limitations (Guo et al., 2025). Training inefficiency significantly reduces its potential benefits. This is clearly demonstrated in Fig. 4 when the x-axis is changed from training steps to GPU hours. However, SPRO effectively addresses the training inefficiency, keeping the dual-model framework comparable to outcome-supervised methods.

As shown in Fig. 4a, our method reduces the additional computational overhead. Due to hardware limitations, our experiments are conducted on 7B models. However, SPRO demonstrates considerable potential for industrial-scale implementation, as its self-guided process rewards inherently benefit from stronger policy models.

## 4    RELATED WORK

Due to page limitations, the discussion of related work is provided in Appendix B.

## 5    CONCLUSION

In this work, we introduced Self-Guided Process Reward Optimization (SPRO), a novel and scalable RL framework for LLMs that eliminates the dependency on costly PRMs while preserving the simplicity of outcome-supervised RL. By introducing Cumulative Process Reward (CPR) as a surrogate for self-guided process signals and proposing Masked Step Advantage (MSA), our method enables rigorous step-level advantage estimation through shared-prompt comparisons. Experiments demonstrate that SPRO significantly improves both accuracy and training efficiency. In particular, it addresses two critical challenges in RL for LLMs: (1) token efficiency, where SPRO reduces reasoning sequence lengths while achieving higher task accuracy, and (2) policy entropy collapse, where our method maintains higher entropy levels, promoting more efficient exploration and mitigating reward hacking during training. The scalability and ease of deployment of SPRO make it particularly suitable for industrial implementation, offering a practical and effective alternative to traditional process reward approaches.

## 6 REPRODUCIBILITY STATEMENT

The source code to reproduce the experimental results is provided in the anonymous repository https://anonymous.4open.science/r/spro. Full implementation details can be found in Appendix C to enable the replication of our work.

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

# A  PRELIMINARIES

In this section, we first introduce the token-level MDP for large language models, along with some definitions of reward and objective function in reinforcement learning.

## A.1  TOKEN-LEVEL MDP FOR LARGE LANGUAGE MODELS

Following with Rafailov et al. (2024), Yuan et al. (2025) and Cui et al. (2025a), the token-level MDP is defined as a tuple $\mathcal{M} = (\mathcal{S}, \mathcal{A}, f, r, \rho)$.

- The action space $\mathcal{A}$ consists of the token vocabulary of any given large language model, while the state space $\mathcal{S}$ comprises all sub-sequences during the inference process. For example, state $\mathbf{s}_t$ at timestep $t$ could be represented as $\mathbf{s}_t = (\mathbf{x}, \mathbf{y}_{<t})$, where $\mathbf{x}$ is the initial input (prompt) and $\mathbf{y}_{<t}$ is the sequence of tokens generated up to step $t - 1$.
- $f(\mathbf{s}_t, \mathbf{a}_t)$ represents a state transition model that updates the state $\mathbf{s}_{t+1}$ by concatenating the newly generated token $\mathbf{a}_t$ to $\mathbf{s}_t$. Formally, this can be expressed as $\mathbf{s}_{t+1} = f(\mathbf{s}_t, a_t)$.
- $\rho(\mathbf{s}_t)$ represents a state distribution constraint that limits the sampling range for each state $\mathbf{s}_t$.
- $r(\mathbf{s}_t, \mathbf{a}_t)$ denotes the token-level reward given after the model outputs token $\mathbf{a}_t$ with input state $\mathbf{s}_t$.

## A.2  MAXIMUM ENTROPY REINFORCEMENT LEARNING IN THE TOKEN-LEVEL MDP

Given a well-defined token-level MDP $\mathcal{M} = (\mathcal{S}, \mathcal{A}, f, r, \rho)$, we can optimize the process reinforcement policy $\pi_\theta$ by using the following entropy-augmented (Williams & Peng, 1991; Ziebart, 2010), KL-constrained objective (Ouyang et al., 2022; Rafailov et al., 2024):

$$\max_{\pi_\theta} \mathbb{E}_{\substack{\mathbf{s}_0 \sim \rho(\mathbf{s}_0), \\ \mathbf{a}_t \sim \pi_\theta(\cdot|\mathbf{s}_t)}} \left[ \sum_{t=0}^{T} \left( \underbrace{(r(\mathbf{s}_t, \mathbf{a}_t)}_{\text{token reward}} + \underbrace{\beta \log \pi_{\text{ref}}(\mathbf{a}_t|\mathbf{s}_t)}_{\text{KL penalty}} \right) + \underbrace{\beta \, \mathcal{H}(\pi_\theta)}_{\text{entropy}} \right]. \quad (9)$$

As mentioned in Rafailov et al. (2024); Ziebart (2010), in the general maximum entropy RL setting, the fixed point solution of Eq. (9) is given as:

$$\pi^*(\mathbf{a}_t|\mathbf{s}_t) = e^{(Q^*(\mathbf{s}_t, \mathbf{a}_t) - V^*(\mathbf{s}_t))/\beta}, \quad (10)$$

where $\pi^*(\mathbf{a}|\mathbf{s})$ is the optimal policy and $Q^*(\mathbf{s}, \mathbf{a})$ is the corresponding optimal soft $Q$-function. The optimal value function $V^*$ is defined as:

$$V^*(\mathbf{s}_t) = \beta \log \sum_{\mathbf{a} \in \mathcal{A}} e^{Q^*(\mathbf{s}_t, \mathbf{a})/\beta}. \quad (11)$$

As shown in Eq. (10), the relationship between the reward function $r(\mathbf{s}, \mathbf{a})$ and optimal policy $\pi(\mathbf{a}|\mathbf{s})$ is not a direct mapping. Instead, the policy is expressed through $Q$-function and V-function, which themselves represent estimates of total future returns.

To further investigate the reward-policy relationship, Rafailov et al. (2024) introduced a modified bellman equation between reward function and value functions using KL-divergence penalty, where:

$$Q^*(\mathbf{s}_t, \mathbf{a}_t) = \begin{cases} r(\mathbf{s}_t, \mathbf{a}_t) + \beta \log \pi_{\text{ref}}(\mathbf{a}_t|\mathbf{s}_t) + V^*(\mathbf{s}_{t+1}), & \text{if } \mathbf{s}_{t+1} \text{ is not terminal,} \\ r(\mathbf{s}_t, \mathbf{a}_t) + \beta \log \pi_{\text{ref}}(\mathbf{a}_t|\mathbf{s}_t), & \text{if } \mathbf{s}_{t+1} \text{ is terminal.} \end{cases} \quad (12)$$

It should be noted that some prior work (Garg et al., 2021; Hejna et al., 2024) has also proposed similar definitions, but they required an assumption that the discount factor $\gamma < 1$. Rafailov et al. (2024) further proved that the relationship in Eq. (12) is indeed one-to-one in the token MDP under mild assumptions, which means there is a **bijection** between reward functions and corresponding optimal $Q$-functions in the token-level MDP.

By log-linearizing the optimal policy fixed point in Eq. (10)

$$\beta \log \pi^*(\mathbf{a}_t|\mathbf{s}_t) = Q^*(\mathbf{s}_t, \mathbf{a}_t) - V^*(\mathbf{s}_t). \quad (13)$$

Substituting in the Bellman equation from Eq. (12) (Nachum et al., 2017; Watson et al., 2023), we have the following function:

$$r(\mathbf{s}_t, \mathbf{a}_t) + V^*(\mathbf{s}_{t+1}) - V^*(\mathbf{s}_t) = \beta \log \frac{\pi^*(\mathbf{a}_t|\mathbf{s}_t)}{\pi_{\text{ref}}(\mathbf{a}_t|\mathbf{s}_t)}. \quad (14)$$

This establishes a mathematical relationship between the process reward function and the policy model.

## B    RELATED WORK

**From RLHF to RLVR.**    In the LLM domain, RL is initially employed to align the model outputs with human preference, a paradigm known as Reinforcement Learning with Human Feedback (RLHF) (Ouyang et al., 2022; Bai et al., 2022). As the focus shifts from alignment to enhancing model reasoning abilities in domains such as math and code, Reinforcement Learning with Verifiable Rewards (RLVR) emerges as a new trend. In this setting, outcome correctness can be verified against ground truth. During the transition, REINFORCE-style algorithms (Shao et al., 2024; Ahmadian et al., 2024; Liu et al., 2025; Williams, 1992; Yu et al., 2025; Hu, 2025), which rely on Monte Carlo estimator of trajectory returns instead of value functions as used in PPO (Schulman et al., 2017), have demonstrated superior efficiency and performance. However, most existing RLVR methods remain outcome-supervised, limiting their ability to provide well-defined process rewards during training.

**Explicit Rewards.**    Uesato et al. (2022) and Lightman et al. (2023) show that PRMs outperform ORMs on reasoning tasks. Building on this, subsequent works (Li et al., 2023; Yu et al., 2023; Zhu et al., 2023; Ma et al., 2023; Jiao et al., 2024; Havrilla et al., 2024) train PRMs to either verify reasoning steps or predict the final return, enabling improved trajectory collection through reranking or filtering. Other works (Feng et al., 2023; Zhang et al., 2024; Chen et al., 2024b) train PRMs for tree-search.

**Implicit Rewards.**    Implicit rewards are broadly adopted in LLM alignment (Rafailov et al., 2024; Ethayarajh et al., 2024; Zhou et al., 2024; Chen et al., 2024a). Rafailov et al. (2024) show that DPO implicitly learns a Q-function. Yuan et al. (2025) extend this idea to Cross Entropy objective. Zhou et al. (2024) use dense implicit rewards for beam search. Cui et al. (2025a) and Zhong et al. (2024) train standalone models to generate implicit rewards for RL training.

## C    EXPERIMENTAL SETUP

**Base model and Dataset.**    We adopt Eurus-2-7B-SFT (Cui et al., 2025a) as our base model, which is fine-tuned from Qwen2.5-Math-7B-Base (Qwen et al., 2025) on mathematical and programming tasks. The RL dataset is Eurus-2-RL-Data (Cui et al., 2025a), which contains math problems ranging from the high school level to International Mathematical Olympiad competition questions, as well as programming tasks primarily at the competitive programming level.

**Resources and Hyper-parameters.**    All experiments are conducted on a single node equipped with 8x NVIDIA A800 GPUs (80G memory) using the veRL framework (Sheng et al., 2025). For optimization, we use the AdamW optimizer with a cosine decay learning rate schedule, initialized at $1 \times 10^{-6}$. For rollout stage, we collect 256 prompts with an oversampling factor of 2 and generate 4 responses per prompt. We apply an accuracy filtering threshold between 0.2 and 0.8, and prompts that fall within this range are prioritized. For training, the batch size is 256 and the micro batch size is 16. The KL coefficient is set to 0, and the entropy coefficient is set to 0.001 for all experiments.

**Outcome Verifiers.**    We employ rule-based reward modeling for outcome verification, aligning with common practices in the recent literature (Guo et al., 2025; Cui et al., 2025a; Lambert et al., 2024). Specifically, we assign a score of 1 if a math answer matches the ground truth and 0 otherwise. For code generation task, we compute the reward as the proportion of passed test cases.

**Evaluation Benchmarks.**    We evaluate models on AIME 2024 (Li et al., 2024), AMC (Li et al., 2024), MATH-500 (Hendrycks et al., 2021), Minerva Math (Lewkowycz et al., 2022), Olympiad-Bench (He et al., 2024), LeetCode (Guo et al., 2024), and LiveCodeBench(v2) (Jain et al., 2024).

**Baseline Algorithms.**    We use vanilla GRPO (Shao et al., 2024) as the baseline algorithm for outcome-supervised RL training, and the previous SoTA PRIME (Cui et al., 2025a) for process-supervised RL training. We maintain consistency in the base model, training dataset, and key hyperparameters with those employed in our proposed method.

## D  COMPARISON OF TEST ACCURACY

Table 3: **Comparison of test accuracy** between SPRO and other baselines using the same base model and training data. We present pass@1 accuracy scores at step 400 on four math benchmarks and two code benchmarks.

| Methods | AMC | MATH | Olympiad | K12 | CodeForces | CodeContests | Avg. |
|---|---|---|---|---|---|---|---|
| Base Model | 21.1 | 48.1 | 13.9 | 40.0 | 5.2 | 11.3 | 22.8 |
| vanilla GRPO | 23.6 | 51.8 | 20.5 | 48.6 | 28.6 | 28.1 | 33.5 |
| PRIME | 31.2 | 52.7 | 25.4 | 54.6 | 26.4 | 25.9 | 36.0 |
| **Ours (SPRO)** | **31.9** | **53.6** | **28.2** | **55.0** | **29.4** | **32.1** | **38.4** |

## E  CASE STUDY

---

**Question:**

Simplify $\frac{1+\sqrt{2}}{2+\sqrt{3}}$. Your solution can be converted to the form $A(1 + \sqrt{B}) - (\sqrt{C} + \sqrt{D})$, where $A$, $B$, $C$, and $D$ are positive integers. What is $A + B + C + D$?

---

**Ours (SPRO):**

The given expression is $\frac{1+\sqrt{2}}{2+\sqrt{3}}$. To simplify this, we can rationalize the denominator by multiplying both the numerator and the denominator by the conjugate of the denominator, which is $2 - \sqrt{3}$.

Multiply the numerator and the denominator by the conjugate of the denominator: $\frac{1+\sqrt{2}}{2+\sqrt{3}} \cdot \frac{2-\sqrt{3}}{2-\sqrt{3}} = \frac{(1+\sqrt{2})(2-\sqrt{3})}{(2+\sqrt{3})(2-\sqrt{3})}$.

Expand the numerator: $(1+\sqrt{2})(2-\sqrt{3}) = 2-\sqrt{3}+2\sqrt{2}-\sqrt{2}\sqrt{3} = 2-\sqrt{3}+2\sqrt{2}-\sqrt{6}$.

Expand the denominator: $(2+\sqrt{3})(2-\sqrt{3}) = 2^2 - (\sqrt{3})^2 = 4 - 3 = 1$.

So, the expression simplifies to: $\frac{2-\sqrt{3}+2\sqrt{2}-\sqrt{6}}{1} = 2 - \sqrt{3} + 2\sqrt{2} - \sqrt{6}$.

Rearrange the terms to match the form $A(1 + \sqrt{B}) - (\sqrt{C} + \sqrt{D})$: $2+2\sqrt{2}-\sqrt{3}-\sqrt{6} = 2(1+\sqrt{2}) - (\sqrt{3}+\sqrt{6})$.

Now we have the expression in the desired form, where $A = 2$, $B = 2$, $C = 3$, and $D = 6$. The final expression is $2(1 + \sqrt{2}) - (\sqrt{3} + \sqrt{6})$, and the sum of the constants is $A + B + C + D = 2 + 2 + 3 + 6 = 13$. Therefore, the answer is 13.

**GRPO:**

The problem asks us to simplify the given expression $\frac{1+\sqrt{2}}{2+\sqrt{3}}$ and then convert it into a specific form. We need to identify the key elements, constraints, and objectives. The key elements are the given expression and the target form. The objective is to simplify the expression and then convert it into the required form. To simplify the given expression, we can start by rationalizing the denominator, which means eliminating the radical in the denominator. We can achieve this by multiplying both the numerator and the denominator by the conjugate of the denominator, which is $2 - \sqrt{3}$. This will help us to get rid of the radical in the denominator and simplify the expression. The expression becomes $\frac{(1+\sqrt{2})(2-\sqrt{3})}{(2+\sqrt{3})(2-\sqrt{3})}$.

Now, we can simplify the numerator and the denominator separately. Now, let's simplify the numerator and the denominator separately. The numerator becomes $(1 + \sqrt{2})(2 - \sqrt{3}) = 2 - \sqrt{3} + 2\sqrt{2} - \sqrt{6}$. The denominator becomes $(2 + \sqrt{3})(2 - \sqrt{3}) = 4 - 3 = 1$. So, the simplified expression is $2 - \sqrt{3} + 2\sqrt{2} - \sqrt{6}$.

We can rewrite this expression in the form $A(1 + \sqrt{B}) - (\sqrt{C} + \sqrt{D})$ by rearranging the terms. Let's try to group the terms with the same radical together. We can rewrite the expression $2 - \sqrt{3} + 2\sqrt{2} - \sqrt{6}$ as $2 + 2\sqrt{2} - \sqrt{3} - \sqrt{6}$.

Now, we can factor out a common term from the first two terms: $2(1 + \sqrt{2}) - \sqrt{3} - \sqrt{6}$.

We can see that the expression is now in the form $A(1 + \sqrt{B}) - (\sqrt{C} + \sqrt{D})$, where $A = 2$, $B = 2$, $C = 3$, and $D = 6$.

Therefore, we have successfully converted the expression into the required form. Now, we can calculate the sum $A + B + C + D$ to get the final answer. The given expression $\frac{1+\sqrt{2}}{2+\sqrt{3}}$ can be simplified by rationalizing the denominator and then rearranging the terms to match the required form. After simplifying, we get $2(1 + \sqrt{2}) - \sqrt{3} - \sqrt{6}$, which is in the form $A(1 + \sqrt{B}) - (\sqrt{C} + \sqrt{D})$. Therefore, $A = 2$, $B = 2$, $C = 3$, and $D = 6$. The sum $A + B + C + D$ is $2 + 2 + 3 + 6 = 13$.

13

---

**PRIME:**

The problem asks us to simplify the given expression $\frac{1+\sqrt{2}}{2+\sqrt{3}}$ and then convert it to a specific form. We need to identify the key elements, constraints, and objectives. The key elements are the given expression and the target form. The constraint is that the solution must be in the specified form. The objective is to simplify the expression and find the values of $A$, $B$, $C$, and $D$. To simplify the given expression $\frac{1+\sqrt{2}}{2+\sqrt{3}}$, we can start by rationalizing the denominator. This involves multiplying the numerator and denominator by the conjugate of the denominator to eliminate the radical in the denominator. The conjugate of $2 + \sqrt{3}$ is $2 - \sqrt{3}$. We will multiply the expression by $\frac{2-\sqrt{3}}{2-\sqrt{3}}$, which is equivalent to 1.

The expression can be broken down into the following steps:1. Multiply the numerator and denominator by the conjugate of the denominator: $\frac{1+\sqrt{2}}{2+\sqrt{3}} \cdot \frac{2-\sqrt{3}}{2-\sqrt{3}}$ 2. Simplify the numerator and denominator separately. 3. Combine the simplified numerator and denominator to obtain the simplified expression

We will now perform the multiplication and simplification steps outlined in the previous action.

1. Multiply the numerator and denominator by the conjugate of the denominator: $\frac{1+\sqrt{2}}{2+\sqrt{3}} \cdot \frac{2-\sqrt{3}}{2-\sqrt{3}} = \frac{(1+\sqrt{2})(2-\sqrt{3})}{(2+\sqrt{3})(2-\sqrt{3})}$

2. Simplify the numerator:$(1 + \sqrt{2})(2 - \sqrt{3}) = 2 - \sqrt{3} + 2\sqrt{2} - \sqrt{6}$

3. Simplify the denominator:$(2 + \sqrt{3})(2 - \sqrt{3}) = 2^2 - (\sqrt{3})^2 = 4 - 3 = 1$

4. Combine the simplified numerator and denominator:$\frac{2-\sqrt{3}+2\sqrt{2}-\sqrt{6}}{1} = 2 - \sqrt{3} + 2\sqrt{2} - \sqrt{6}$

We have obtained the simplified expression $2 - \sqrt{3} + 2\sqrt{2} - \sqrt{6}$. Now, we need to convert this expression to the target form $A(1 + \sqrt{B}) - (\sqrt{C} + \sqrt{D})$.

We can start by rearranging the terms to group the constants and the square root terms:$2 - \sqrt{3} + 2\sqrt{2} - \sqrt{6} = (2 + 2\sqrt{2}) - (\sqrt{3} + \sqrt{6})$

We can factor out a common factor from the first group of terms:$2 + 2\sqrt{2} = 2(1 + \sqrt{2})$

Now, we can rewrite the expression in the target form:$2(1 + \sqrt{2}) - (\sqrt{3} + \sqrt{6})$

By comparing this with the target form, we can identify the values of $A$, $B$, $C$, and $D$:

$A = 2$, $B = 2$, $C = 3$, and $D = 6$

We have obtained the values of $A$, $B$, $C$, and $D$ and rewritten the expression in the target form. Now, we need to verify the accuracy of our approach and conclusions.

We will review the steps taken to ensure that the simplification and conversion to the target form are correct.

1. We rationalized the denominator by multiplying by the conjugate, which is a standard and valid technique.

2. We simplified the numerator and denominator correctly.

3. We rearranged the terms and factored out a common factor, which is a valid algebraic manipulation.

4. We compared the expression with the target form and identified the values of $A$, $B$, $C$, and $D$ correctly.

Our approach seems to be valid, and we have not found any potential errors or inconsistencies in our reasoning.

However, we can improve our approach by double-checking our calculations and ensuring that we have considered all possible simplifications and conversions.

We can also evaluate our process by considering what worked well (e.g., using the conjugate to rationalize the denominator) and what could be improved (e.g., double-checking calculations). We were given the task to simplify the expression $\frac{1+\sqrt{2}}{2+\sqrt{3}}$ and convert it to the form $A(1 + \sqrt{B}) - (\sqrt{C} + \sqrt{D})$, where $A$, $B$, $C$, and $D$ are positive integers. We needed to find the values of $A$, $B$, $C$, and $D$ and calculate their sum.

Our thought process involved:

1. Assessing the problem and identifying the key elements, constraints, and objectives.

2. Simplifying the expression by rationalizing the denominator and multiplying by the conjugate.

3. Advancing our reasoning by performing the multiplication and simplification steps.

4. Synthesizing the information by combining the simplified numerator and denominator and converting the expression to the target form.

5. Verifying the accuracy of our approach and conclusions by reviewing the steps and checking for potential errors or inconsistencies.

Our final conclusion is that the simplified expression is $2(1 + \sqrt{2}) - (\sqrt{3} + \sqrt{6})$, which corresponds to the values $A = 2$, $B = 2$, $C = 3$, and $D = 6$.

Therefore, the sum of $A$, $B$, $C$, and $D$ is $2 + 2 + 3 + 6 = 13$.

The final answer is: 13

