# OpenReview forum: "Self-Guided Process Reward Optimization with Redefined Step-wise Advantage for Process Reinforcement Learning"
_ICLR.cc/2026/Conference — Submitted to ICLR 2026_

### Official Review · Reviewer_3UT3 · 2025-10-14

**Soundness:** 1
**Presentation:** 2
**Contribution:** 2
**Rating:** 4
**Confidence:** 4

**Summary:**

The paper proposes to remove the costly process reward model and use the internal signal of the policy model itself for process supervision. It correspondingly proposes Cumulative Process Rewards (CPR) and Masked Step Advantage to calculate the advantage of each step. The empirical experiments validate its high efficiency.

**Strengths:**

- Empirically show that process signals intrinsically derived from the policy itself benefit the performance.

- Propose Cumulative Process Rewards (CPR) and Masked Step Advantage to obtain the step advantage.

- Empirical experiments validate the high efficiency of SRPO.

**Weaknesses:**

> **Overclaim the theoretical contribution**
>---
>- Although authors claim they theoretically demonstrate that process rewards can be derived intrinsically from the policy model itself, the proposition 1 in section 2.1 is not novel. The similar proofs can be found in [1][2]
>
>- The derivations of Cumulative Process Reward (CPR) are also not novel, which is exactly the same as [3]
>
>[1] Zhong, Han, et al. "DPO Meets PPO: Reinforced Token Optimization for RLHF." Forty-second International Conference on Machine Learning.
>
>[2] Rafailov, Rafael, et al. "From $ r $ to $ Q^* $: Your Language Model is Secretly a Q-Function." First Conference on Language Modeling.
>
>[3] Xie, Tengyang, et al. "Exploratory Preference Optimization: Harnessing Implicit Q*-Approximation for Sample-Efficient RLHF." The Thirteenth International Conference on Learning Representations.


> **Unfair comparison**
> ---
> As shown in Fig.1, Fig.4, **the training appears not to have fully converged.** All the training has stopped early. Hence, although SPRO displays an early performance surge, it does not necessarily lead to final superior performance.


> **Doubtful claim about length bias**
> ---
>- The authors claim MSA does not induce the length bias. However, their argument is more like a step-number bias, i.e., MSA would not make the policy model prefer responses with more steps. Although there are definitely some correlations between the step number and response length. However, can they really be equivalent? Moreover, there is a length normalization $1/|y|$ in the SRPO loss, which is proven to be a factor inducing the length bias [4].
>- According to Table 2, the response length of SPRO across all datasets has a weirdly sharp decrease, which shows a contrary length bias towards shorter response length.
>
>[4]Liu, Zichen, et al. "Understanding r1-zero-like training: A critical perspective." arXiv preprint arXiv:2503.20783 (2025).



> **Typos**
> ---
> - L23: vaniila -> vanilla
> - L96: advantages estimation -> advantage estimation
> - L248: implict -> implicit
> - L304: 3th -> third
> - L459: its -> their

**Questions:**

> Zero rewards for longest responses

As demonstrated in L304-308, if only the third response contains a valid step, the advantage of this step is zero. Therefore, for each group of responses, there must be some steps of the longest response unlearnt. Is it a waste of computation? Why does CPR and MSA can benefit this trade-off?

> Is outcome supervision always necessary for the training?

Current training signals combine the outcome and process supervision, which is understandable since when the policy model is not strong enough during the early training stage, the policy needs outcome rewards to improve its performance. However, in the later training stage, whether the pure process signals can make an effect is doubtful. Does CPR always necessarily combine the outcome rewards to work?

> Contrary observation of entropy change

As shown in Fig.5(b), the entropy of vanilla GRPO remains stable, which is contrary to the previous empirical studies [5]

[5] Cui, Ganqu, et al. "The entropy mechanism of reinforcement learning for reasoning language models." arXiv preprint arXiv:2505.22617 (2025).

> Ablation of MSA

As shown in Fig.2(a), PRIME uses a different advantage calculation method, which aggregates all process rewards into a single group. Is grouping rewards at the same step (MSA) really superior to grouping rewards among all steps in the SPRO framework?  Does the advantage calculation of PRIME also work well in the SPRO framework?

---

> ### Author Response · Authors · 2025-11-24
>
> Thank you for your detailed and constructive feedback, as well as for pointing out the typos. Please see our detailed responses to your comments and suggestions below. We hope that our responses could address your concerns. Please let us know if you have further questions.
>
> > **Weakness 1**: "**Overclaim the theoretical contribution** - Although authors claim they theoretically demonstrate that process rewards can be derived intrinsically from the policy model itself, the proposition 1 in section 2.1 is not novel. The similar proofs can be found in [1,2]- The derivations of Cumulative Process Reward (CPR) are also not novel, which is exactly the same as [3]"
>
> **Reply**:
> Thank you for raising this point. Firstly, we would like to emphasize that, **as stated in our summarized main contributions (Lines 149–159) and as also recognized by multiple reviewers**, our main contributions are as follows:
> - propose the efficient PRL framework SPRO,
> - redefine step-level advantage by CPR and MSA,
> - empirically demonstrate the effectiveness of SPRO
>
> We have revised the wording in the updated version (LINE 19-20), and this does not compromise our main contributions as listed above.
>
> Secondly, we would like to clarify that although there may be similar proofs in [1,2], **their theorems/propositions can not be directly applied to our subsequent methods.
> The differences are as follows**:
> -  In both [1] and [2], the implicit rewards are derived from _preference models_ and _preference data_, and the policy is optimized under a preference-model objective. In contrast, our proposition generalizes to **any LLM** and renders the implicit rewards **independent of the specific training objective used to obtain the LLM**, which forms the foundation of our self-guided framework.
> - **The purpose of CPR is fundamentally different from the trajectory coverability in [3]**. We introduce CPR to align with the cumulative nature of LLM representations and to provide the reward signals used in the subsequent MSA procedure. In contrast, Xie et al. [3] focus on online sampling and use trajectory coverability to derive sample complexity bounds. Although the mathematical summation forms may look similar, the underlying conceptual motivation and their roles within the respective frameworks are distinct.
>
> [1] Zhong, Han, et al. "DPO Meets PPO: Reinforced Token Optimization for RLHF." Forty-second International Conference on Machine Learning.
>
> [2] Rafailov, Rafael, et al. "From  to : Your Language Model is Secretly a Q-Function." First Conference on Language Modeling.
>
> [3] Xie, Tengyang, et al. "Exploratory Preference Optimization: Harnessing Implicit Q*-Approximation for Sample-Efficient RLHF." The Thirteenth International Conference on Learning Representations.
>
>
> > **Weakness 2**:"Unfair comparison. As shown in Fig.1, Fig.4, **the training appears not to have fully converged.** All the training has stopped early. Hence, although SPRO displays an early performance surge, it does not necessarily lead to final superior performance."
>
> **Reply**: This is a good question. We would like to clarify that **the observed early stopping reflects practical constraints shared across RL works [1,2,3,4,5], rather than an unfair comparison**. In fact, full convergence of RL training is difficult to achieve, and the experiments in DAPO[1], GSPO[2], Dr. GRPO[3], RLOO[4] and PRIME[5] are all early stopped for this reason.
>
> Furthermore, a central goal of SPRO is to improve PRL training efficiency under realistic computational budgets. In such settings, **final superior performance is meaningful only when compared under comparable compute**. A method that requires six times more computational resources to obtain marginally higher accuracy is substantially less practical in real-world training.
>
> [1] DAPO: An Open-Source LLM Reinforcement Learning System at Scale.
>
> [2] Group Sequence Policy Optimization.
>
> [3] Understanding R1-Zero-Like Training: A Critical Perspective.
>
> [4] Back to Basics: Revisiting REINFORCE Style Optimization for Learning from Human Feedback in LLMs.
>
> [5] Process Reinforcement Through  Implicit Rewards.

---

> ### Author Response · Authors · 2025-11-24
>
> > **Weakness 3**:"Doubtful claim about length bias. -The authors claim MSA does not induce the length bias. However, their argument is more like a step-number bias, i.e., MSA would not make the policy model prefer responses with more steps. Although there are definitely some correlations between the step number and response length. However, can they really be equivalent? Moreover, there is a length normalization  in the SRPO loss, which is proven to be a factor inducing the length bias [4]. - According to Table 2, the response length of SPRO across all datasets has a weirdly sharp decrease, which shows a contrary length bias towards shorter response length."
>
> **Reply**: This is a good question. We would like to clarify that:
> - In our work, **a step is defined at the token level**; therefore, the number of steps exactly equals to the response length.
> - Shorter response length lies in the motivation of process reinforcement learning and token efficiency is also what we pursue. Thus, **it is expected rather than weird**.
>
>
> > **Weakness 4**: "Typos"
>
> **Reply**:Thank you for pointing them out. We have corrected them in the updated submission.
> > **Q1**: "Zero rewards for longest responses.As demonstrated in L304-308, if only the third response contains a valid step, the advantage of this step is zero. Therefore, for each group of responses, there must be some steps of the longest response unlearnt. Is it a waste of computation? Why does CPR and MSA can benefit this trade-off?"
>
> **Reply**: This is a good question. In implementation, **the computations of CPR and MSA are quite trivial**. We simply reuse the intermediate variables already stored during the outcome-supervised RL process to compute them. Therefore, the additional cost is negligible and cannot be regarded as a real trade-off.
>
> > **Q2**: "Is outcome supervision always necessary for the training? Current training signals combine the outcome and process supervision, which is understandable since when the policy model is not strong enough during the early training stage, the policy needs outcome rewards to improve its performance. However, in the later training stage, whether the pure process signals can make an effect is doubtful. Does CPR always necessarily combine the outcome rewards to work?"
>
> **Reply**: This is a great question. Since our focus is on providing process rewards under outcome-supervised RL, outcome supervision is always present in our current setting. However, we agree that it would be an interesting direction for future research to investigate what happens if outcome supervision is removed, in which the training would be fully self-supervised, and to determine when such removal would be most appropriate.
>
>
> > **Q3**: "Contrary observation of entropy change. As shown in Fig.5(b), the entropy of vanilla GRPO remains stable, which is contrary to the previous empirical studies [5]"
>
> **Reply**: **The observation of vanilla GRPO is not contrary**. As shown in Figure 9 of Cui _et al._’s paper, the authors also note that “setting the coefficient at 0.005 successfully stabilizes policy entropy.” In our experiments, we adopt the default coefficient value, which coincidentally corresponds to the stable setting in our case as well.
>
> > **Q4**: "Ablation of MSA. As shown in Fig.2(a), PRIME uses a different advantage calculation method, which aggregates all process rewards into a single group. Is grouping rewards at the same step (MSA) really superior to grouping rewards among all steps in the SPRO framework? Does the advantage calculation of PRIME also work well in the SPRO framework?"
>
> **Reply**: This is a great question. We conducted ablation studies to examine the role of MSA in the SPRO framework. The Avg. Test Acc is shown below:
>
> | Step        | 120   | 160   | 400   | 412       | 596   |
> | ----------- | ----- | ----- | ----- | --------- | ----- |
> | **CPR w/o MSA** | 0.338 | 0.358 | 0.384 | **0.111** | –     |
> | **CPR w/ MSA**  | 0.338 | 0.354 | 0.387 | 0.385     | 0.399 |
>
> It's evidenced that grouping all rewards among all steps (the advantage calculation of PRIME) in the SPRO framework makes the policy gradient optimization unstable.

---

> ### Author Response · Authors · 2025-11-27
>
> Dear Reviewer 3UT3,
>
> Thank you very much for the time and effort you dedicated to reviewing our work and for the valuable comments you provided. We carefully considered all your feedback, addressed the concerns in our response, and revised our paper accordingly.
>
> We sincerely hope that our clarifications and additional experiments help resolve your concerns. If you have any further concerns or questions, please let us know and we are more than happy to further address them.
>
> Thank you again for your thoughtful and constructive review.
>
> Best regards,
>
> The Authors

---

### Official Review · Reviewer_5xtC · 2025-10-31

**Soundness:** 3
**Presentation:** 3
**Contribution:** 3
**Rating:** 6
**Confidence:** 2

**Summary:**

This paper proposes a new reward formulation called Cumulative Process Reward (CPR), which leverages the policy model’s own output as a source of reward.
Based on this idea, the authors introduce a new advantage estimation method, MSA, demonstrating improvements in both reasoning performance and training efficiency.
Theoretical analysis further shows that the method can mitigate reward hacking while preserving sufficient exploration and maintaining high policy entropy.

Parts of this review were discussed with a colleague to ensure clarity and accuracy.

**Strengths:**

1. The paper clearly articulates its contributions and presents a coherent logical progression. It first introduces the CPR concept, builds upon it to propose the MSA module, and finally develops the overall SPRO algorithmic framework. The overall structure is well-organized, and the presentation is smooth and easy to follow.
2. The experimental design convincingly demonstrates that the proposed SPRO algorithm outperforms prior baselines in terms of both accuracy and training efficiency.
3. The theoretical derivation of CPR is clear.

**Weaknesses:**

1. One of the main contributions of this paper is the introduction of the self-guided reward (CPR), which is theoretically formulated as a general reward mechanism that appears not to be constrained by reasoning tasks or step-wise processes. However, the experiments only evaluate CPR within reasoning-oriented benchmarks. It would significantly strengthen the paper to include results demonstrating CPR’s generality—specifically, whether it can serve as a replacement for reward models in other common RL fine-tuning scenarios beyond reasoning tasks.
2. The quantitative experiments are conducted on a single base model (Eurus-2-7B-SFT). To better demonstrate the robustness and general applicability of the proposed method, it would be beneficial to include experiments on multiple model scales.

**Questions:**

See weakness.

---

> ### Author Response · Authors · 2025-11-24
>
> > **W1**: "One of the main contributions of this paper is the introduction of the self-guided reward (CPR), which is theoretically formulated as a general reward mechanism that appears not to be constrained by reasoning tasks or step-wise processes. However, the experiments only evaluate CPR within reasoning-oriented benchmarks. It would significantly strengthen the paper to include results demonstrating CPR’s generality—specifically, whether it can serve as a replacement for reward models in other common RL fine-tuning scenarios beyond reasoning tasks."
>
> **Reply**: This is an insightful question, and adaptation to general scenarios is indeed important. We are working on evaluating SPRO under common RL fine-tuning scenarios and will report the results as soon as they are available.
>
>
> > **W2**: "The quantitative experiments are conducted on a single base model (Eurus-2-7B-SFT). To better demonstrate the robustness and general applicability of the proposed method, it would be beneficial to include experiments on multiple model scales."
>
> **Reply**: Thank you for your valuable suggestions. To demonstrate that SPRO generalizes beyond a single base model, we conducted additional experiments on **Qwen2.5-14B-Instruct**. SPRO consistently improves performance across training steps.
>
> The Avg. Test Acc is shown below:
>
> | Step                     | 0     | 32    | 64    | 128   | 200   |
> | ------------------------ | ----- | ----- | ----- | ----- | ----- |
> | **Qwen2.5-14B-Instruct** | 0.382 | 0.422 | 0.442 | 0.439 | 0.458 |
> | **Eurus-2-7B-SFT**       | 0.227 | 0.239 | 0.277 | 0.336 | 0.358 |
>
> These results indicate that SPRO scales to larger models and continues to provide stable improvements, suggesting robustness and general applicability.

---

> ### Author Response · Authors · 2025-11-27
>
> Dear Reviewer 5xtC,
>
> Thank you very much for the time and effort you dedicated to reviewing our work and for the valuable comments you provided. We carefully considered all your feedback, addressed the concerns in our response, and revised our paper accordingly.
>
> We sincerely hope that our clarifications and additional experiments help resolve your concerns. If you have any further concerns or questions, please let us know and we are more than happy to further address them.
>
> Thank you again for your thoughtful and constructive review.
>
> Best regards,
>
> The Authors

---

### Official Review · Reviewer_cqQc · 2025-10-31

**Soundness:** 2
**Presentation:** 1
**Contribution:** 2
**Rating:** 2
**Confidence:** 3

**Summary:**

The paper proposes SPRO, a process-aware RL method for reasoning LLMs that removes the extra PRM by deriving a self-guided token-level signal from the policy itself and redefines step-wise advantage via two constructs: Cumulative Process Reward and Masked Step Advantage (MSA). CPR is obtained by telescoping a policy–value identity to produce a per-prefix cumulative log-ratio against a reference model; MSA normalizes CPR across trajectories at the same step within a shared-prompt group, and the sum of (normalized) outcome reward and MSA is used in a PPO-style clipped objective (Algorithm 1). Experiments on math and code benchmarks report better accuracy and shorter outputs vs. GRPO and PRIME, with lower GPU hours.

**Strengths:**

SPRO avoids an auxiliary PRM and computes process feedback directly from the policy and a fixed reference model, preserving a dual-model training footprint.

MSA enforces per-step comparisons within a prompt’s rollouts, which directly targets the length-bias problem common in outcome-only grouping. The construction and “masked mean” baseline are clearly spelled out.

Simulation gains include shorter trajectories, higher accuracy, and better entropy than PRIME/GRPO under their setup, with tables/figures that are easy to read.

**Weaknesses:**

The key identity used to define CPR (Eq. 2–5) is derived by combining the max-entropy fixed-point and a Bellman-style relation that holds for the optimal policy/value. In practice, SPRO replaces $\pi^*$ with the current $\pi_\theta$​ to compute the log-ratio sum. The paper does not quantify the bias introduced by this substitution nor provide a bound that links CPR to true per-step advantage under model mismatch. This is central because CPR then drives the update.

Proposition 1 asserts that “any LLM is an optimal soft Q-function for some reward,” by scaling logits to define Q and then defining V accordingly so that \pi matches a soft-optimal policy (Eq. 1). This is a definition-level construction; it does not imply that the induced reward (or the resulting token-credit signal) is useful or less biased/noisier than an external PRM. The additional claim that “stronger LLMs provide more accurate credit assignment” is also not supported here.

The derivation of CPR and the log-ratio uses a KL-regularized, entropy-augmented setting with inverse temperature \beta and a KL term to the reference policy. But in experiments, the KL coefficient is set to 0, while CPR still sums log-ratios vs. \pi_ref. This disconnect raises questions: if the training does not penalize deviation from \pi_ref, why reference the right anchor for process signal, and what sets \beta numerically?


The paper attributes entropy rises and reduced length to SPRO’s fine-grained feedback, but these are correlational observations. No ablation shows that removing CPR or changing MSA grouping eliminates the effect; no statistical tests or confidence bands are reported.

**Questions:**

The definition of CPR relies on the fixed-point relation of the optimal entropy-regularized policy.
How valid is this identity when used with the current non-optimal policy \pi_\theta? Specifically, can you quantify or bound the bias introduced when replacing \pi^* with \pi_\theta? Without this, why should CPR correlate with the true per-step advantage?

Proposition 1 shows that any policy can be seen as optimal for some constructed reward.
What ensures that the “implied reward” derived from logits is meaningful for credit assignment rather than an arbitrary scaling?
Have you empirically verified that CPR correlates with external process rewards (e.g., from a PRM)?


Your derivation of CPR assumes a KL-regularized objective with temperature β and reference policy, but experiments disable the KL penalty. How do you justify computing log-ratios vs. \pi_ref. When the actual optimization omits that regularization?
Does \beta still have a numerical role during training? If not, is CPR just a scaled log-probability difference?

You claim SPRO enables “self-guided” credit assignment without a PRM. Is the guidance truly intrinsic, or is it implicitly dependent on the chosen reference \pi_ref? What happens if \pi_ref  is very weak or unrelated to  \pi_\theta?
Does CPR degenerate to pure likelihood training in that case?

---

> ### Author Response · Authors · 2025-11-24
>
> Thank you for the thoughtful comments and we appreciate the feedback you have given us. We have addressed your points below and are happy to provide any further clarifications.
>
> > **W1**: "The key identity used to define CPR (Eq. 2–5) is derived by max entropy fix point ... that holds for the optimal policy/value. SPRO replace $\pi^*$ with $\pi_{\theta}$, without quantifying the bias introduced by this substitution nor provide a bound."
>
> **Reply**: This is a good question. We would like to clarify that **we do not introduce bias by utilizing the current $\pi_{\theta}$**, since $\pi_{\theta}$ is the optimal soft Q-function for the corresponding reward function $r(s_t, a_t)$, as shown in Eq.(2-5). And CPR is defined through an implicit reward function, which is the reward of the fix point optimal policy in Eq.(9).   (Proposition 1 addresses this issue, indicating that any LLM can be regarded as an optimal policy and there always exists a corresponding implicit reward function $r(s_t, a_t)$. Under this view, the notion of bias does not occur.)
>
> In addition, we would like to **restate our main ideas** in detail. First, in Appendix.A, we show the relationship between an optimal policy and its corresponding implicit reward function under MaxEnt RL. We then regard the current training policy checkpoint as an optimal policy, which induces an implicit reward function. Building upon this, we define CPR. Through derivation, it can be strictly expressed in the form of $V(S_0) + \sum(.)$ , as shown in Eq.(5) . However, because we subtract a mean baseline while computing the MSA, the $V(S_0)$ term cancels out. This means that our defined CPR is equivalent to optimizing by the form $\sum(.)$ without $V(S_0)$. Both formulations yield the same policy gradient and optimal solution (Definition 1 in [1]; Definition 1 in [2]).
>
> [1] Direct Preference Optimization: Your Language Model is Secretly a Reward Model
>
> [2]From r to Q∗: Your Language Model is Secretly a Q-Function
>
> > **W2**: Proposition 1 asserts that “any LLM is an optimal soft Q-function for some reward,” by scaling logits to define Q and then defining V accordingly so that \pi matches a soft-optimal policy (Eq. 1). This is a definition-level construction; it does not imply that the induced reward (or the resulting token-credit signal) is useful or less biased/noisier than an external PRM. The additional claim that “stronger LLMs provide more accurate credit assignment” is also not supported here.
>
> **Reply**:
> Thank you for your thoughtful comments.
> 1. **The notion of implicit rewards is well established in prior works** [1,2]; broadly adopted  in LLM alignment [3,4,5]; please refer to Figure 1 in [1] for an illustration of implicit rewards for credit assignment. Our experimental results also show that the derived CPR (corresponding to some reward function) serves as good signals for process reinforcement learning. Note that **we do not claim that an implicit reward function is “less biased” or “less noisy” than an external PRM**. This position is not taken in our paper. To the best of our knowledge, there is also no gold-standard explicit reward function (i.e. a PRM) that can serve as a definitive reference for correlation.
> 2. Since any LLM is the optimal soft Q-functions for some reward functions in the token-level MDP[2], better downstream task performance implies that the corresponding reward function is more aligned with the task, which enables the LLM to perform more accurate credit assignment. We have clarified this point in the revised version; please refer to Lines 209-211.
>
> [1] From r to Q∗: Your Language Model is Secretly a Q-Function
>
> [2] Direct Preference Optimization: Your Language Model is Secretly a Reward Model
>
> [3] Weak-to-strong search: Align large language models via searching over small language models
>
> [4] Dpo meets ppo: Reinforced token optimization for rlhf
>
> [5] Free process rewards without process labels
>
> > **W3**: The derivation of CPR and the log-ratio uses a KL-regularized, entropy-augmented setting with inverse temperature \beta and a KL term to the reference policy. But in experiments, the KL coefficient is set to 0, while CPR still sums log-ratios vs. \pi_ref. This disconnect raises questions: if the training does not penalize deviation from \pi_ref, why reference the right anchor for process signal, and what sets \beta numerically?
>
> **Reply**: This is a good question. We would like to clarify that **our implementation does penalize deviation from $\pi_{ref}$** , fully consistent with Eq. (9). The confusion arises from an implementation detail in veRL (also present in other RL frameworks): **there are two possible places to apply the KL penalty**, one in KL loss and the other in KL-in-reward. In our implementation, we set the KL coefficient in the KL loss to zero, which means that the KL loss is disabled and the KL penalty is applied through the reward. Therefore, we do penalize deviation from $\pi_{ref}$ and it is aligned with Eq.(9).

---

> ### Author Response · Authors · 2025-11-24
>
> > **W4**: The paper attributes entropy rises and reduced length to SPRO’s fine-grained feedback, but these are correlational observations. No ablation shows that removing CPR or changing MSA grouping eliminates the effect; no statistical tests or confidence bands are reported.
>
> **Reply**: Thank you for your valuable suggestions. We conducted ablation studies to examine the role of CPR and MSA. When CPR is removed, we utilize $r = \beta\log \frac{\pi_\theta}{\pi_{\mathrm{ref}}}$ , following prior works [1,2].
>
> **Ablation Results.**  Removing CPR or MSA causes policy optimization to collapse or become unstable. The Avg. Test Acc is shown below:
>
> | Step             | 120   | 160       | 400   | 412       | 596   |
> | ---------------- | ----- | --------- | ----- | --------- | ----- |
> | **w/o CPR, w/o MSA**| 0.356 | **0.114** | –     | –         | –     |
> | **w/ CPR, w/o MSA**  | 0.338 | 0.358     | 0.384 | **0.111** | –     |
> | **w/ CPR, w/ MSA**   | 0.338 | 0.354     | 0.387 | 0.385     | 0.399 |
>
> As shown, removing either CPR or MSA causes the training collapse or unstable, which leads to erratic and uninformative entropy and response-length trajectories. The ablation study shows that, with CPR and MSA, the policy remains stable, together with the entropy rise and length shortening phenomenon, suggesting that these components are causally responsible for the observed effects rather than merely correlational.
>
> [1]From r to Q∗: Your Language Model is Secretly a Q-Function.
>
> [2]Direct Preference Optimization: Your Language Model is Secretly a Reward Model.
>
>
> > **Q1**: "The definition of CPR relies on the fixed-point relation of the optimal entropy-regularized policy. How valid is this identity when used with the current non-optimal policy $\pi_\theta$? Specifically, can you quantify or bound the bias introduced when replacing $\pi^*$ with $\pi_\theta$? Without this, why should CPR correlate with the true per-step advantage?"
>
> **Reply**: Please refer to our response to W1.
>
> > **Q2**: "Proposition 1 shows that any policy can be seen as optimal for some constructed reward. What ensures that the “implied reward” derived from logits is meaningful for credit assignment rather than an arbitrary scaling? Have you empirically verified that CPR correlates with external process rewards (e.g., from a PRM)?"
>
> **Reply**: Please refer to our response to W2.
>
>
> > **Q3**: "Your derivation of CPR assumes a KL-regularized objective with temperature β and reference policy, but experiments disable the KL penalty. How do you justify computing log-ratios vs. \pi_ref. When the actual optimization omits that regularization? Does \beta still have a numerical role during training? If not, is CPR just a scaled log-probability difference?"
>
> **Reply**: Please refer to our response to W3.
>
> > **Q4**: "You claim SPRO enables “self-guided” credit assignment without a PRM. Is the guidance truly intrinsic, or is it implicitly dependent on the chosen reference $\pi_{ref}$? What happens if $\pi_{ref}$ is very weak or unrelated to $\pi_\theta$? Does CPR degenerate to pure likelihood training in that case?"
>
> **Reply**: This is a good question. **Firstly, $\pi_{ref}$ can't be unrelated to $\pi_\theta$ or arbitrarily weak.** The reference model is the initial point of the policy model in RL common practice, therefore it could not be “unrelated” to $\pi_{\theta}$ . Moreover, it should not be too “weak”. If the trajectories sampled from a weak model are all negative samples, the resulting gradient for policy updates would vanish, which defeats the fundamental purpose of policy gradient optimization. **Secondly, the guidance is intrinsic.** Since the ref model represents the starting point of the policy, it doesn't compromise the intrinsic nature of the guidance mechanism.

---

> ### Author Response · Authors · 2025-11-27
>
> Dear Reviewer cqQc,
>
> Thank you very much for the time and effort you dedicated to reviewing our work and for the valuable comments you provided. We carefully considered all your feedback, addressed the concerns in our response, and revised our paper accordingly.
>
> We sincerely hope that our clarifications and additional experiments help resolve your concerns. If you have any further concerns or questions, please let us know and we are more than happy to further address them.
>
> Thank you again for your thoughtful and constructive review.
>
> Best regards,
>
> The Authors

---

> > ### Comment · Reviewer_cqQc · 2025-11-28
> >
> > I thank the authors for their detailed response; most of my main concerns have been addressed. One possible improvement over the current algorithm is that the reference policy may not have enough coverage in many cases, such that $\pi_{\text{ref}}(a \vert s)$ is close to zero. In this case, having $\pi_{\text{ref}}$ in the denominator may lead to numerical errors or instability during training. It may be helpful to include additional experiments that consider different levels of reference policies. I have raised my rating based on the current version.

---

> > > ### Comment · Reviewer_cqQc · 2025-11-28
> > >
> > > I am not sure what happened with openreview, but it currently does not allow me to change the rating. I am leaving a note here and will update the score as soon as this issue is resolved.

---

> ### Author Response · Authors · 2025-11-28
>
> Thank you for carefully reading our response and revised paper and we are sincerely grateful for your decision to raise the score. We greatly appreciate your constructive comments and clarifying questions, which helped improve the paper. We will continue working on the important issues you raised (particularly regarding the $\pi_{ref}$ ). Thank you again!

---

### Official Review · Reviewer_NH4D · 2025-11-02

**Soundness:** 3
**Presentation:** 3
**Contribution:** 3
**Rating:** 8
**Confidence:** 4

**Summary:**

The paper introduces Self-Guided Process Reward Optimization (SPRO), a new reinforcement learning framework for LLM reasoning that removes the need for explicit Process Reward Models (PRMs).
Specifically, SPRO proposed two algorithm improvements, Cumulative Process Reward (CPR) and Masked Step Advantage (MSA), which is a self-derived process signal from the policy model’s own logits, serving as an intrinsic token-level reward and a step-wise advantage estimator computed within shared-prompt groups, enabling fair per-step comparisons without length bias.
Experiments on math and code benchmarks show 17.5% accuracy gain and 3.4× higher training efficiency than GRPO, while using no additional computational cost. SPRO also maintains high policy entropy and shorter reasoning traces, mitigating reward hacking and improving token efficiency.

**Strengths:**

1. The paper clearly identifies the computational inefficiency of PRM-based methods and proposes a conceptually elegant self-guided alternative grounded in theory (policy-as-Q-function perspective).
2. The derivation connecting token-level MDPs, policy logits, and process rewards (Eq. 1–5) is rigorous and builds well on prior implicit reward literature (e.g., DPO/PRIME).
3. The presentation of CPR + MSA, especially Fig. 2–3, offers an intuitive and well-structured comparison with GRPO/PRIME.
SPRO consistently outperforms strong baselines on diverse math/code tasks and demonstrates efficiency improvements of up to 6.7× in GPU hours.
4. Maintaining a dual-model (policy + reference) setup without an extra PRM makes deployment convenient and affordable for large-scale reasoning models.

**Weaknesses:**

1. Results are restricted to 7B-scale models; scalability claims to larger models reasoning are not empirically validated.
2. While the “policy-as-reward-model” proposition is appealing, it risks circular reasoning—reward quality depends on policy quality, which itself evolves via those rewards.
3. Baselinses are mainly include GRPO and PRIME; missing other RL methods (e.g., Rest-MCTS*, Reinforce++, DAPO) weakens generality claims.

**Questions:**

1. Could the method extend naturally to multi-turn dialogue or tool-use reasoning, where “steps” are semantically heterogeneous?

---

> ### Author Response · Authors · 2025-11-24
>
> All authors are grateful for your time devoted to reviewing this paper and for your positive feedback about our work. Please see our detailed responses to your comments and suggestions below.
>
> > **W1**: "Results are restricted to 7B-scale models; scalability claims to larger models reasoning are not empirically validated."
>
> **Reply**: Thank you for your valuable suggestions. To empirically validate that SPRO scales beyond a single 7B model, we conducted additional experiments on Qwen2.5-14B-Instruct. SPRO consistently improves performance across training steps.
>
> The Avg. Test Acc is shown below:
>
> | Step                     | 0     | 32    | 64    | 128   | 200   |
> | ------------------------ | ----- | ----- | ----- | ----- | ----- |
> | **Qwen2.5-14B-Instruct** | 0.382 | 0.422 | 0.442 | 0.439 | 0.458 |
> | **Eurus-2-7B-SFT**       | 0.227 | 0.239 | 0.277 | 0.336 | 0.358 |
>
> These results show that SPRO continues to provide performance gains at the 14B scale, indicating that the method generalizes reliably to larger models and supports our scalability claims.
>
> > **W2**: "While the “policy-as-reward-model” proposition is appealing, it risks circular reasoning, reward quality depends on policy quality, which itself evolves via those rewards."
>
> **Reply**: We appreciate your point! Since the outcome signal supervises the entire procedure, the reward generation is anchored in outcome supervision rather than being solely dependent on the evolving policy, thereby avoiding a vicious cycle of circular reasoning.
>
> > **W3**: "Baselinses are mainly include GRPO and PRIME; missing other RL methods (e.g., Rest-MCTS*, Reinforce++, DAPO) ."
>
> **Reply**: This is a good question. Note that our work primarily focuses on _online_ RL, so we do not include SFT methods such as Rest-MCTS* as baselines. Among outcome-supervised algorithms, GRPO is the most representative baseline, while Reinforce++ and DAPO are both variants derived from GRPO.
> > **Q1**: "Could the method extend naturally to multi-turn dialogue or tool-use reasoning, where “steps” are semantically heterogeneous?"
>
> **Reply**: Yes, the method can be naturally extended to multi-turn dialogue or tool-use reasoning. Since our token-level steps represent the finest granularity, aggregating rewards over a subset of steps would accommodate such scenarios. However, defining appropriate step boundaries in these semantically heterogeneous contexts remains a major challenge. In addition, the length and structure of steps may vary across trajectories, which would require further adaptation of the grouping strategy. Thank you for your insightful question; we believe it points to a valuable direction for future research.

---

### Meta-Review · Area_Chair_MeSX · 2026-01-08

**Summary:**

This paper proposes SPRO, a process reinforcement learning framework that derives process rewards intrinsically from the policy model itself, eliminating the need for costly auxiliary PRMs. The key technical contributions are Cumulative Process Reward (CPR) and Masked Step Advantage (MSA) for step-wise advantage estimation.

During the review period, the concerns regarding this paper were centered on the novelty and experimental scope. I am also very concerned about the novelty. The core idea of this paper has been introduced for nearly a decade [1,2] in classical RL, and has been intensively studied over the past year (as heavily cited in the paper and rebuttal). I would suggest the authors refactor the structure of this paper and highlight which part is known and which part is new—for example, Section 2.2 seems to have already been presented in previous papers [3]. Otherwise, it is very hard to assess the real contribution of this paper.

Given the current state of this paper, I don't think it is ready to be published at ICLR.

[1] Equivalence Between Policy Gradients and Soft Q-Learning
[2] Bridging the Gap Between Value and Policy Based Reinforcement Learning
[3] Efficient (Soft) Q-Learning for Text Generation with Limited Good Data

**Reviewer Concerns:**

Addressed concerns: many empirical questions, including scalability, more ablations, and KL coefficient misunderstanding. Outstanding concerns: mainly novelty and early stopping. My concerns on novelty are stated above. As for the claim on the necessity of early stopping, I don't fully agree, as many recent works are able to prolong RL until convergence [1]. But I would treat the early stopping issue as minor, as it is still quite open for now.

[1] Stabilizing Reinforcement Learning with LLMs: Formulation and Practices

**Reviewer Scores:**

Reviewer NH4D = 8, Reviewer cqQc likely 2 -> 4, Reviewer 5xtC = 6, Reviewer 3UT3 = 4

---

### Decision · Program_Chairs · 2026-01-26

Reject